# A Retrospective Cohort Study on the Side Effects of Intrathecal Morphine Administration Combined with General Anaesthesia Versus General Anaesthesia Alone in Prostatectomy Patients

**DOI:** 10.3390/medicina61040732

**Published:** 2025-04-15

**Authors:** Timon Marvin Schnabel, Katharina Fetz, Hanaa Baagil, Kim Kutun, Claus Eisenberger, Mark Ulrich Gerbershagen

**Affiliations:** 1Department of Anaesthesiology, Witten/Herdecke University, Cologne—Holweide Hospital, Neufelder Str. 32, 51067 Cologne, Germanykutunk@kliniken-koeln.de (K.K.);; 2Institute for Research in Operative Medicine (IFOM), Witten/Herdecke University, Ostmerheimerstraße 200, 51109 Cologne, Germany; katharina.fetz@uni-wh.de; 3Department of Anaesthesiology and Operative Intensive Care, Cologne Merheim Medical Centre, Ostmerheimerstraße 200, 51109 Cologne, Germany; 4Department of General Surgery, Witten/Herdecke University, Cologne Hospital, Neufelder Str. 32, 51067 Cologne, Germany; eisenbergercf@kliniken-koeln.de

**Keywords:** analgesic consumption, general anaesthesia, intrathecal morphine, numerical rating scale (NRS), pain management, postoperative pain, prostatectomy, retrospective cohort study, side effects, spinal anaesthesia

## Abstract

*Background and Objectives:* Prostatectomy is a common surgical procedure for prostate cancer, the most frequently diagnosed cancer in the male population. The choice of anaesthetic technique has a significant impact on postoperative pain management. The changes in recommendations between 2015 and 2021 prompted this study to evaluate the impact of intrathecal morphine administration in combination with general anaesthesia compared to general anaesthesia alone on postoperative analgesic consumption and the associated side effects. *Material and Methods:* A single-centre retrospective cohort study was conducted, analysing data from 202 patients who underwent a prostatectomy between 2015 and 2021. Patients were divided into two groups: 147 patients received intrathecal morphine combined with general anaesthesia, while 49 patients received general anaesthesia alone. Key postoperative parameters, including numerical rating scale (NRS) scores, analgesic consumption, and side effects (e.g., nausea, pruritus, hypotension, and respiratory depression) were evaluated. Statistical analyses were performed using Mann–Whitney U-tests and multiple regression models. *Results:* The group receiving intrathecal morphine showed a significant decrease in NRS pain scores at rest and during movement in the recovery room (*p* < 0.001). The need for postoperative analgesics, especially opioids such as piritramide, was reduced in this group. No significant increase in serious side effects such as respiratory depression was observed. *Conclusions:* The present study investigates the potential of intrathecal morphine combined with general anaesthesia as a promising approach to improve pain management in prostatectomy patients. By reducing pain intensity, this method shows significant clinical benefits. In addition, the absence of a significant increase in serious adverse events reinforces the safety of this approach. However, further studies are warranted to assess the long-term outcomes and explore optimal dosing strategies. The reintroduction of this anaesthetic technique has great potential to improve patient recovery and satisfaction following major surgery.

## 1. Introduction

Prostate cancer is the most commonly diagnosed malignancy in men and a leading cause of cancer-related mortality worldwide [1,2]. In Germany, prostatectomy is a commonly performed surgical procedure, with 21,850 cases documented in 2016 alone [3]. The rising prevalence of prostate cancer has been attributed to the increasing use of prostate-specific antigen (PSA) screening, a practice that has improved early detection but also raised concerns about overdiagnosis [4,5,6].

Effective postoperative pain management is critical to improving recovery after a prostatectomy [7,8]. Inadequate management of postoperative pain can impede mobilisation, prolong healing, and increase the risk of complications, including deep vein thrombosis and pulmonary embolism [9,10]. Spinal anaesthesia, consisting of the local anaesthetics bupivacaine and morphine in combination with general anaesthesia (GA), has been adopted as standard practice in many institutions [11], including the Cologne Hospital, Holweide, Germany, with the aim of improving pain management.

Although the analgesic benefits of intrathecal morphine (ITM) are well-documented, its use is not without risk. Adverse effects such as pruritus, nausea, and respiratory depression have been reported [12,13]. In addition, international guidelines differ in their recommendations for the use of ITM in prostatectomy due to these potential complications. For example, the European Association of Urology (EAU) guidelines emphasise the importance of balancing pain management with minimising side effects [4]. Similarly, the PROSPECT guidelines, as established by Lemoine et al. in 2021 [7], conversely advise against its use due to the unresolved issue of its side effects and the unstudied duration of its effect [7]. While Joshi et al. (2015) [14] highlighted the efficacy of ITM in reducing postoperative pain, they also emphasised the need for caution regarding its associated side effects, particularly in the context of prostatectomy. In contrast to Lemoine et al. (2021), Joshi et al. (2015) recommended ITM for prostatectomy, citing its analgesic effect and the low side effects when applied correctly [7,14]. Enhanced recovery after surgery protocols emphasise the need for multimodal approaches to optimise patient outcomes while minimising risks [15]. These protocols highlight the potential of combining GA with ITM to achieve superior postoperative pain management and accelerated recovery, as intrathecal administration of morphine provides prolonged analgesic effects in prostatectomy patients [15,16].

This study investigates the efficacy and safety of ITM in combination with GA for prostatectomy. Specifically, it compares this approach to GA alone, focusing on postoperative pain management, analgesic consumption, and the incidence of side effects. The results of this study are intended to provide evidence to guide clinical decision-making and improve patient care.

## 2. Materials and Methods

This retrospective, single-centre study was conducted at Cologne-Holweide Hospital, Germany, with the objective of evaluating the efficacy and safety of ITM in combination with GA for patients undergoing open radical prostatectomy. Ethical approval was granted by the Ethics Committee of the University of Witten/Herdecke (approval number S-240/2024), and the study adhered to the Declaration of Helsinki and the European Union’s General Data Protection Regulation (GDPR). Data collection and analysis were conducted between October 2024 and January 2025, including patients treated between 2015 and 2021.

Patients were eligible for inclusion in the study if they had a histologically confirmed diagnosis of prostate cancer and underwent open radical prostatectomy, with complete and consistent documentation in their medical records. The exclusion criteria included incomplete data or additional surgical procedures unrelated to prostate cancer. A total of 202 patients met the inclusion criteria.

Patient data were extracted from both electronic and paper records archived in the hospital’s CCP^®^ system. The extracted variables included:Demographics: age, height, and weight;Preoperative parameters: comorbidities, numerical rating scale (NRS) chronic pain score, prior opioid use, and home medication;Intraoperative parameters: anaesthetic agents, hemodynamic monitoring, and fluid and blood product administration;Postoperative parameters: pain scores, analgesic requirements, complications, ICU/IMC stay, hospitalisation duration, urinary output, and haematuria.

The anaesthetic protocol patients were allocated into two groups:GA Group: received GA alone;GA + ITM Group: received ITM with 0.3 mg morphine combined with bupivacaine under strict aseptic conditions, in addition to GA.

The induction of general anaesthesia was facilitated by the administration of propofol, sufentanil or fentanyl and cisatracurium. The maintenance of anaesthesia was achieved using sevoflurane or desflurane, supplemented as necessary by intravenous agents. Intraoperative haemodynamic monitoring included non-invasive and invasive blood pressure measurement, electrocardiography, and pulse oximetry. Fluid management was guided by estimated blood loss and anaemia (haemoglobin < 8 g/dL), with crystalloids and blood products administered accordingly. Vasopressors (cafedrine/theodrenaline or noradrenaline) were used as needed to maintain haemodynamic stability. Furthermore, prophylactic dexamethasone (2–8 mg) was administered to prevent postoperative nausea and vomiting.

Patients were monitored in the recovery room for vital signs, respiratory function, and pain management. Pain intensity was assessed at rest and during mobilisation using the NRS. The analgesic regimens included intravenous piritramide, metamizole, and oral oxycodone, with naloxone administered as required to counter opioid-related side effects. Nausea and vomiting were treated with dimenhydrinate, ondansetron, metoclopramide, or promethazine. Postoperative complications, including pruritus, respiratory depression, haemodynamic instability, and surgical complications, were documented. Patients were transferred to the IMC or ICU based on clinical needs.

Data analysis was performed using SPSS version 29 (IBM). Descriptive statistics were used to summarise the patient characteristics. The Mann–Whitney U test was applied to compare the non-parametric variables between groups. Multiple linear regression analysis was conducted to identify predictors of postoperative pain and analgesic consumption. A *p*-value of < 0.05 was considered statistically significant. If necessary, Pearson’s r was used to evaluate the degree of correlation between variables [17].

The anonymised dataset and detailed protocols for data collection and analysis are available upon request to facilitate replication and further research.

## 3. Results

### 3.1. Demographic Data

In the period between 2015 and 2021, 202 patients underwent radical prostatectomies at Cologne-Holweide Hospital. Of these patients, 147 received ITM alongside GA, 49 received GA alone, and in 6 cases, the anaesthetic technique was unclear.

The mean age of the GA + ITM group was found to be 68.18 ± 7.70 years, in comparison to 70.27 ± 6.98 years in the GA-only group. No significant differences were observed between the two groups with respect to body weight and height, with mean weights of 85.64 ± 17.04 kg (GA + ITM) and 84.29 ± 13.99 kg (GA-only). The mean height in the GA + ITM group was recorded as 175.91 cm. The duration of hospitalisation was found to be shorter in the GA + ITM group (12.41 ± 4.67 days) in comparison to the GA-only group (14.37 ± 7.86 days), but not significant. The duration of ICU/IMC stay was comparable, with an average of 0.94 ± 0.75 days in the GA + ITM group and 0.87 ± 0.65 days in the GA-only group. Table 1 presents demographic data of the ITM and No-ITM groups, showing no statistically significant differences in age, weight, height, hospital stay, or ICU/IMC stay.

### 3.2. Pain Assessment and Analgesic Administration

The assessment of pain was conducted utilising the NRS in the recovery room (RR) and on the IMC/ICU. The analysis encompassed data from 138 patients in the ITM + GA group and 43 patients in the GA-only group. However, 21 patients were excluded on the basis of missing values.

#### Recovery Room Pain Assessment and Medication Administration

Patients in the ITM + GA group reported lower pain scores in the RR (NRS at rest: M = 1.05, Mdn = 0.00; during movement: M = 1.22, Mdn = 0.00) compared to the GA-only group (rest: M = 2.20, Mdn = 3.00; movement: M = 2.18, Mdn = 2.00).

Analgesic administration was lower in the ITM group (0.22 vs. 0.67 medications), with piritramide (1.18 mg vs. 3.62 mg), metamizole (0.04 g vs. 0.05 g), and oxycodone (0.07 mg vs. 0.29 mg) being the most frequently administered analgesics. The median time to first analgesic administration was similar (4 h 15 min vs. 4 h 30 min).

The Mann–Whitney U test confirmed a significant difference in NRS scores at rest (*p* < 0.001) but no significant difference during movement (*p* = 0.055). Figure 1 shows boxplots of NRS scores at rest and during movement in the RR, comparing ITM and No-ITM groups.

A multiple linear regression analysis was conducted to examine the predictors of NRS scores at rest in the RR. The model was found to be statistically significant (F(14, 138) = 2.223, *p* = 0.009), with an adjusted R^2^ value of 10.1%, which explained 18.4% of the variance.

ITM was found to be associated with significantly lower NRS scores (b = −1.156, *p* = 0.001). Additionally, age was found to have a negative correlation with pain scores (b = −0.069, *p* = 0.002), and fentanyl administration was linked to reduced pain (b = −1.843, *p* = 0.039).

The remaining factors, including weight, propofol, midazolam, thiopental, sufentanil, piritramide, comorbidities (hypertension, diabetes, hypothyroidism, COPD), and time elapsed since ITM administration, were found to have no significant effect (all *p* > 0.05). Table 2 displays the results of the multiple linear regression analysis identifying predictors of NRS scores at rest in the recovery room.

Table 3 presents the results of a multiple linear regression analysis identifying factors influencing NRS scores at movement in the RR. The model was statistically significant (F(14, 138) = 1.791, *p* = 0.045) and explained 15.4% of the variance (adjusted R^2^ = 6.8%).

ITM was significantly associated with lower NRS scores during movement (b = −0.872, *p* = 0.028). Age also showed a negative correlation with pain scores (b = −0.082, *p* = 0.001). Other factors, including weight, midazolam, thiopental, sufentanil, piritramide, comorbidities (hypertension, diabetes, hypothyroidism, COPD), and time elapsed since ITM administration, had no significant effect (all *p* > 0.05).

The ITM group demonstrated a reduced requirement for analgesics when compared to the GA-only group, with a mean piritramide dose of 1.18 mg (SD = 2.76) in the ITM group, as opposed to 3.62 mg (SD = 4.50) in the GA-only group. Furthermore, the total number of analgesics administered was lower in the ITM group (M = 0.22, SD = 0.45) than in the GA-only group (M = 0.67, SD = 0.61). Intravenous morphine was not required in the ITM group, whereas the GA-only group received an average of 0.12 mg (SD = 0.76). Figure 2 shows boxplots comparing (a) intravenous piritramide doses and (b) the number of analgesics administered in the RR by ITM.

The Mann–Whitney U test confirmed significant differences in piritramide dosage (*p* < 0.001) and the total number of analgesics administered (*p* < 0.001).

Two multiple linear regression models analysed the factors influencing piritramide dosage and the number of analgesics administered in the RR. Neither model reached statistical significance (piritramide dosage: F(14, 129) = 1.466, *p* = 0.133, R^2^ = 0.137; analgesic count: F(14, 129) = 1.223, *p* = 0.266, R^2^ = 0.117). No significant predictors were identified (all *p* > 0.05), though sufentanil (*p* = 0.083), weight (*p* = 0.099), propofol dosage (*p* = 0.077), and arterial hypertension (*p* = 0.086) approached significance for the number of administered analgesics. Full regression results are provided in Section A.1 and Section A.2.

A comparison of the pain measurement and analgesic administration timing between the ITM (N = 136) and non-ITM (N = 42) groups was conducted, revealing no significant difference (ITM: 4:22 h ± 0:52; non-ITM: 4:32 h ± 1:09; *p* = 0.309).

A pain assessment (NRS) was conducted at rest and during movement in the IMC/ICU, revealing no significant differences between the groups (all *p* = 0.05). The ITM group exhibited lower mean NRS scores at rest (0.77 vs. 1.24) and during movement (0.90 vs. 1.72). However, these differences did not attain statistical significance (*p* = 0.05 and *p* = 0.05).

The utilisation of analgesics was comparable between the two groups, with the ITM group receiving a mean of 0.92 analgesics, and the GA-only group receiving 0.95 in the IMC/ICU (*p* = 0.89). No statistically significant differences were observed for the specific analgesic dosages (all *p* > 0.05), including piritramide (*p* = 0.095), metamizole (*p* = 0.990), and oxycodone (*p* = 0.438).

The timing of analgesic administration was also comparable between groups (ITM: 11:30 h, GA-only: 12:54 h, *p* = 0.452).

In summary, the results of the Mann–Whitney U test demonstrate that there are no statistically significant differences between the treatment and control groups with regard to the NRS scores, the total number of analgesics administered, or the dosages of specific analgesics on the IMC/ICU ward.

### 3.3. Complications and Side Effects

#### 3.3.1. Haemodynamic Complications

The haemodynamic effects of ITM were assessed through heart rate, blood pressure, and circulatory support medication at multiple time points up to 24 h post-intervention. Heart rate showed no significant differences between the groups at any time (all *p* > 0.4). Systolic blood pressure was significantly higher in the ITM group at 5 and 10 min post-intervention (*p* = 0.012, *p* = 0.004), while diastolic pressure showed a transient increase at 5 min (*p* = 0.049). A regression analysis indicated a significant negative association between ITM and both systolic (*p* = 0.015) and diastolic (*p* = 0.004) blood pressure at 30 min, with intraoperative atropine also contributing to lower pressures. Other factors showed no significant influence (all *p* > 0.2). Full regression details can be found in Section A.3 and Section A.4.

Intraoperative use of Akrinor^®^ showed no significant difference between groups (*p* = 0.321), while atropine was administered at a lower dose in the ITM group (*p* = 0.003). Conversely, noradrenaline use was higher in the ITM group (*p* = 0.003). A multiple linear regression model explained only 1.9% of the variance in noradrenaline administration and was not statistically significant (*p* = 0.861). No significant associations were found between noradrenaline dosage and any predictor, including ITM, patient characteristics, comorbidities, or other medications (all *p* > 0.3). The full results are available in Section A.5.

The ITM group had a longer mean time for circulatory medication administration (22.52 min vs. 11.98 min), but the difference was not statistically significant (*p* = 0.121).

#### 3.3.2. Respiratory Complications

Oxygen saturation was measured in three stages: preoperatively, during ITM administration, and at intervals up to 24 h post-intervention. As the data were collected over time, the sample size decreased.

In the GA-only group, mean oxygen saturation ranged from 97.80% preoperatively to a peak of 99.00% at 30 min before declining to 96.00% at 6 h and 97.25% at 24 h. The ITM group exhibited a comparable trend, commencing at 97.42%, reaching a peak of 99.77% at 30 min, and decreasing to 96.67% at 6 h and 96.06% at 24 h. Figure 3 provides a visual representation of the mean saturation levels of both groups over the course of this 24-h period.

The Mann–Whitney U test revealed no statistically significant differences in oxygen saturation between the groups at any time point (e.g., preoperative: *p* = 0.455; 30 min: *p* = 0.722).

The analysis of oxygen supplementation was conducted at 6 and 12 h following the intervention. It was observed that both groups exhibited a mean flow rate of 1.55 L/min at 6 h (median = 2.00 L/min). At the 12 h mark, the GA-only group received 1.10 L/min (Mdn. = 1.00 L/min), while the ITM group received 1.35 L/min (Mdn. = 2.00 L/min). Subsequent analysis revealed no significant differences at either time point (6 h: *p* = 0.653; 12 h: *p* = 0.664).

#### 3.3.3. Hospital Stay

The Mann–Whitney U test showed no significant difference in hospital stay between the ITM (M = 12.41 days, SD = 4.67) and non-ITM patients (M = 14.37 days, SD = 7.86; *p* = 0.107).

A multiple linear regression model was employed to analyse the predictors of hospital stay duration, explaining 17.4% of the variance (R^2^ = 0.174, adjusted R^2^ = 0.114; F(10, 138) = 2.9, *p* = 0.002). Diabetes mellitus was found to be significantly associated with prolonged hospital stays (+5.26 days, *p* < 0.001). ITM was associated with a reduced stay (−1.74 days), though this effect was not significant (*p* = 0.069). Weight exhibited a positive, yet non-significant, association (*p* = 0.086). The remaining variables, encompassing age, hypertension, intraoperative noradrenaline utilisation, pain scores, analgesic timing, and transfer duration, demonstrated no significant impact (all *p* > 0.1). The complete results of the study are provided in Section A.6.

#### 3.3.4. Effect of Intrathecal Morphine on Postoperative Nausea and Vomiting

Postoperative nausea and vomiting (PONV) were assessed in 178 patients (ITM: N = 135, GA-only group: N = 43). In the RR, the mean incidence of emesis was 0.00 in the GA-only group and 0.01 in the ITM group, while nausea occurred at rates of 0.05 and 0.06, respectively. The anti-emetic medications utilised were comparable, with dimenhydrinate being the most prevalent agent administered (GA-only: 10.07 mg; ITM: 9.00 mg). No statistically significant differences were observed for either emesis (*p* = 0.423) or nausea (*p* = 0.760) in the RR.

In the IMC/ICU ward, no significant differences were observed in nausea incidence (*p* = 0.130) or timing (*p* = 0.401). Similarly, emesis occurrence (*p* = 0.816) and timing (*p* = 0.506) showed no significant variation. Furthermore, anti-emetic doses, including dimenhydrinate (*p* = 0.689) and ondansetron (*p* = 0.570), were found to be comparable, with a slight but non-significant trend observed for promethazine (*p* = 0.078).

A multiple linear regression model explained 29.1% of the variance in emesis in the RR (R^2^ = 0.291, adjusted R^2^ = 0.236; F(11, 141) = 5.269, *p* < 0.001). The analysis revealed that the only significant variable was nausea in the RR (*p* < 0.001). However, ITM did not demonstrate significance (*p* = 0.335). The complete results are presented in Section A.6.

A separate model for nausea in the RR (ITM, fentanyl, sufentanil, dexamethasone, age, weight, pain scores, analgesic timing, and diabetes) was not statistically significant (F(10, 142) = 0.793, *p* = 0.635, R^2^ = 0.053, adjusted R^2^ = −0.014). ITM demonstrated no significant association with nausea (b = 0.058, *p* = 0.211), nor did other variables, including dexamethasone (*p* = 0.512), sufentanil (*p* = 0.894), pain scores, age, weight, or diabetes. Full regression details can be found in Section A.7.

#### 3.3.5. Other Complications

The following section will analyse the complications listed in the discharge summary.

Among 195 patients with documented complications, 50% had no complications (ITM: 49%, non-ITM: 53%). The most frequent complications were:Vesicoureteral anastomotic insufficiency (paravasation): 29% (ITM: 31%, non-ITM: 24%);Lymphoceles: 7% (ITM: 6%, non-ITM: 8%);Wound dehiscence: 3% (ITM: 3%, non-ITM: 4%);Postoperative delirium: 1% (ITM: 1%, non-ITM: 0%);Mortality: 1.5% overall (non-ITM: 6%, ITM: 0%).

The cause of death in three patients was documented as cardiac arrest.

The Mann–Whitney U test revealed no statistically significant differences between the ITM and non-ITM groups with regard to the majority of complications, including vesicoureteral anastomotic insufficiency (*p* = 0.4 00), wound dehiscence (*p* = 0.651), and postoperative delirium (*p* = 0.651). The proportion of patients with no complications was also comparable (*p* = 0.651). Figure 4 displays a bar chart showing the distribution of side effects reported in discharge summaries, grouped by ITM administration.

## 4. Discussion

This study demonstrates that ITM significantly reduces postoperative pain and opioid consumption in patients undergoing radical prostatectomy. Patients receiving ITM exhibited lower NRS scores at rest and during movement in the immediate postoperative period, required fewer additional analgesics, and exhibited no significant increase in major complications compared to the GA-only group. While transient haemodynamic changes were observed, no sustained haemodynamic instability, respiratory depression, or increased rates of PONV were detected.

The findings of this study confirm the efficacy of ITM in reducing postoperative pain intensity and opioid consumption, which is consistent with prior research [18,19,20,21,22,23,24,25,26]. Patients in the ITM group reported significantly lower NRS scores at rest and during movement in the immediate postoperative period, mirroring earlier studies that highlight the ability of ITM to delay the time to first analgesic request and decrease overall opioid requirements [27,28,29]. Joshi et al. (2015) and Lemoine et al. (2021) advocate for neuraxial opioids as part of multimodal analgesic strategies [7,14], a recommendation reinforced by the present study. However, while Joshi et al. recommended ITM for prostatectomy in 2015, Lemoine et al. (2021) later advised against its routine use due to concerns over side effects and unclear long-term benefits. The 0.3 mg dose used in this study exceeds current guideline recommendations of 0.1–0.2 mg [7,11], aligning instead with older studies that prioritised analgesic efficacy over minimising side effects.

Despite robust pain relief in the RR, no significant differences were observed in the pain scores or analgesic consumption in the IMC/ICU, suggesting a diminishing effect of ITM beyond the initial 12–24 h. This aligns with the prior research showing that ITM’s analgesic benefits are most pronounced within the first postoperative day [1,3,11]. The prolonged action of ITM, due to its hydrophilic properties and slow clearance from the cerebrospinal fluid, supports its role in early postoperative pain management [16,18] but also reinforces the importance of determining the lowest effective dose to minimise side effects.

The haemodynamic changes following ITM administration were minimal, with transient increases in systolic and diastolic blood pressure at 5 and 10 min post-intervention. These findings are consistent with reports by Joshi et al. (2015), Lemoine et al. (2021), and Ummenhofer et al. (2000), which indicate that neuraxial opioids may induce short-term blood pressure fluctuations but do not cause sustained instability [7,14,16]. Reise and Van Aken (2011) similarly noted transient haemodynamic changes with neuraxial opioids, particularly in combination with local anaesthetics, reinforcing the need for initial monitoring but not prolonged postoperative surveillance [15].

Respiratory complications, particularly delayed respiratory depression, are a well-documented concern with ITM. However, no significant differences were observed between the groups in the oxygen saturation levels or the need for supplemental oxygen, corroborating findings that ITM at doses below 0.3 mg rarely leads to clinically relevant respiratory depression [19,23]. Sultan et al. (2011) and Mugabure Bujedo (2017) emphasise that appropriate dosing and monitoring can effectively mitigate respiratory risks [18,19]. Koning et al. (2022) caution that respiratory depression remains the most severe ITM-related side effect, highlighting the necessity of vigilance in clinical practice [23]. The absence of respiratory depression in this study may reflect its relatively small sample size and the low incidence of this complication reported in large-scale studies [12,29,30,31,32].

The PONV rates were comparable between groups, suggesting that prophylactic anti-emetic strategies were effective in both cohorts. While previous research has identified nausea and vomiting as common side effects of ITM [33,34,35], studies by Rathmell et al. (2005) and Koning et al. 2020 and 2022 highlight the success of multimodal PONV management in reducing symptom severity in patients receiving neuraxial opioids [23,27,30]. Joshi et al. (2015) further emphasise the importance of preventing PONV to improve patient satisfaction and recovery outcomes [14].

Regarding postoperative complications, no significant differences were observed in vesicoureteral anastomotic insufficiency, lymphoceles, or wound dehiscence. This aligns with the findings of Zand et al. (2015), who reported no increase in surgical site complications with ITM use [36]. Paravasation and lymphoceles, which occurred at rates similar to previous studies, are more likely influenced by surgical technique than by the analgesic method [37,38]. The overall safety profile of ITM, as observed in this study, supports its continued use when appropriate monitoring protocols are followed, as recommended by Joshi et al. (2015) [14]. Stein et al. previously described the risk–benefit profile of intrathecal opioids as manageable when integrated into perioperative care [20].

The findings indicate that ITM is an effective analgesic adjunct for radical prostatectomy, providing robust early postoperative pain relief while reducing systemic opioid requirements. The absence of significant respiratory or haemodynamic instability supports its safety when appropriately dosed and monitored. However, given the lack of consensus on the optimal ITM dose, future research should focus on determining the lowest effective dose to balance analgesia and the side effects.

The present study is subject to several limitations, which must be given due consideration. First, the retrospective nature of the study design limits the ability to establish causality and increases the potential for selection bias. Additionally, the single-centre nature of the study implies limitations in terms of its generalisability to other institutions or patient populations. The absence of randomisation further reduces the strength of comparisons between groups, as confounding variables may not have been evenly distributed. The analysis utilised limited measuring points, a methodological decision that inevitably introduces a certain degree of imprecision into the results. The lack of documentation concerning preoperative home medications, which have the capacity to influence postoperative outcomes, introduces an unaccounted source of variability. Additionally, the documentation of urinary output side effects was inadequate, resulting in their exclusion from the statistical analysis. The reduced sample size and unequal distribution between the treatment and GA-only group have been demonstrated to reduce statistical power and limit the robustness of conclusions drawn. Moreover, the evaluation of pruritus, a prevalent side effect of intrathecal morphine, was impeded by the absence of systematic documentation in patient records.

These limitations underscore the necessity for prospective, multicentre, randomised controlled trials to validate these findings and furnish more robust evidence for clinical practice.

## 5. Conclusions

Intrathecal morphine (0.3 mg) has been shown to significantly improve postoperative pain management in prostatectomy, reducing pain intensity and opioid consumption without increasingly severe side effects. This study reinforces the safety and efficacy of ITM but suggests that the 0.3 mg dose may exceed current guideline recommendations, which advocate for lower doses to minimise side effects.

## Figures and Tables

**Figure 1 medicina-61-00732-f001:**
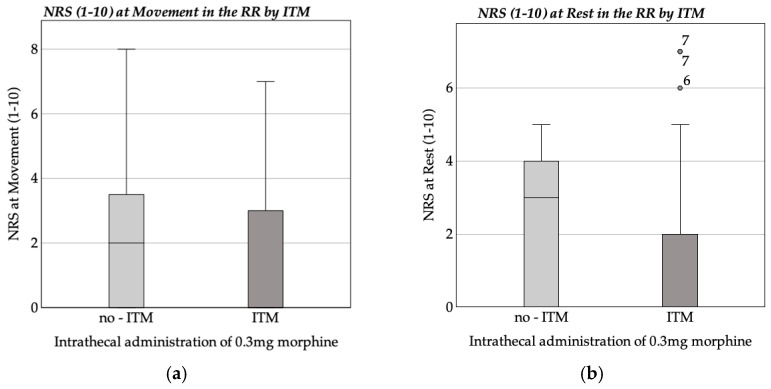
Boxplot of NRS at rest and movement in the recovery room (RR). (**a**) Average NRS at movement in the recovery room by ITM; (**b**) average NRS at rest in the recovery room by ITM.

**Figure 2 medicina-61-00732-f002:**
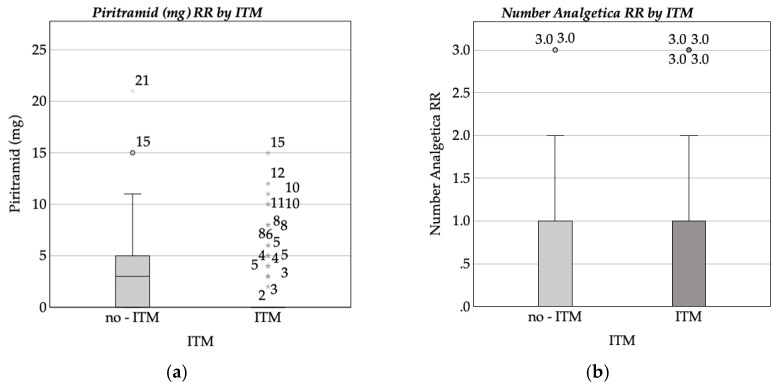
Boxplot of (**a**) intravenous piritramide dose (mg) administered in the recovery room (RR) by intrathecal morphine (ITM) administration status; (**b**) number of analgesics administered in the recovery room (RR) by intrathecal morphine (ITM) administration status.

**Figure 3 medicina-61-00732-f003:**
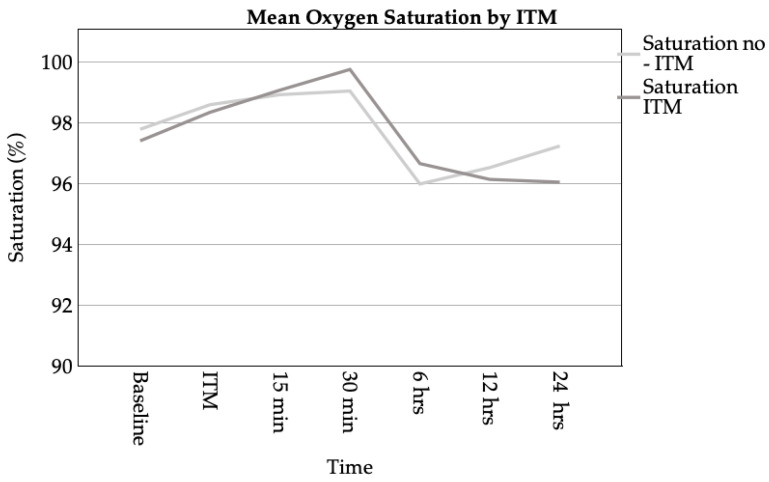
Line chart of mean oxygen saturation over time for patients with and without ITM administration. Measurements were taken at baseline, during ITM administration, and at intervals of 15 min, 30 min, 6 h, 12 h, and 24 h post-intervention.

**Figure 4 medicina-61-00732-f004:**
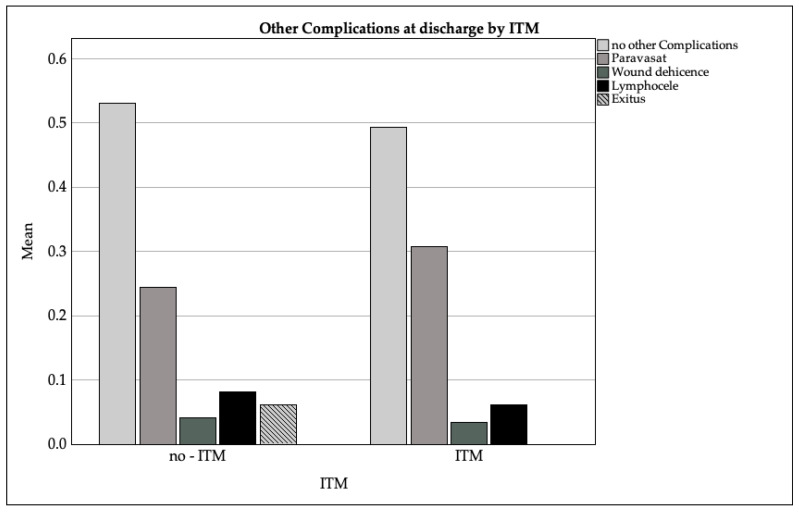
Bar chart of the distribution of side effects reported in the discharge summaries grouped by intrathecal morphine (ITM) administration.

**Table 1 medicina-61-00732-t001:** Demographic data presentation of data as number n or mean value ± standard deviation.

	ITM	No—ITM	*p*-Value
n	147	49	
Age (Years)	68.18 ± 7.697	70.27 ± 6.984	*p* = 0.107
Weight (Kg)	85.64 ± 17.043	84.29 ± 13.995	*p* = 0.760
Height (cm)	175.91 ± 9.485	175.91 ± 8.070	*p* = 0.453
Hospital Stay (days)	12.41 ± 4.668	14.37 ± 7.857	*p* = 0.107
ICU/IMC Stay (days)	0.94 ± 0.751	0.87 ± 0.647	*p* = 0.478

**Table 2 medicina-61-00732-t002:** Results of the multiple linear regression analysis for predictors of NRS scores at rest in the recovery room (N = 153).

NRS (1–10) at Rest in the Recovery Room					
Coefficients	*b*	SE	*β*	*t*	*p*
(Constant)	9.246	1.992		4.641	<0.001
ITM	−1.156	0.345	−0.269	−3.353	0.001
Age	−0.069	0.021	−0.275	−3.223	0.002
Weight	−0.010	0.009	−0.091	−1.076	0.284
Propofol	−0.004	0.002	−0.149	−1.652	0.101
Midazolam	−0.005	0.005	−0.091	−1.144	0.254
Thiopental	−0.010	0.006	−0.128	−1.577	0.117
Sufentanil	−0.082	0.148	−0.048	−0.553	0.581
Piritramide	0.034	0.046	0.059	0.726	0.469
Fentanyl.	−1.843	0.884	−0.168	−2.085	0.039
Arterial Hypertension	−0.079	0.294	−0.022	−0.269	0.788
Diabetes Mellitus	0.181	0.490	0.030	0.369	0.713
Hypothyroidism	−0.386	0.563	−0.055	−0.686	0.494
COPD	−4.85	0.749	−0.056	−0.647	0.518
Time after ITM	−2.217 × 10^−5^	0.000	−0.041	−0.522	0.603

Remarks: N = 153; R^2^ = 0.184; corr. R^2^ = 0.101; F(14, 138) = 2.223; *p* = 0.009. *b* = regression coefficient; *SE* = standard error of the coefficient; *β* = standardised regression coefficient; *t* = *t*-value; *p* = *p*-value; R^2^ = coefficient of determination; F = F-statistic; ITM = intrathecal morphin; COPD = chronic obstructive pulmonary disease.

**Table 3 medicina-61-00732-t003:** Results of the multiple linear regression analysis for predictors of NRS scores at movement in the recovery room (N = 153).

NRS (1–10) at Movement in the Recovery Room					
Coefficients	*b*	SE	*β*	*t*	*p*
(Constant)	9.614	2.275		4.226	<0.001
ITM	−0.872	0.394	−0.181	−2.214	0.028
Age	−0.082	0.025	−0.290	−3.344	0.001
Weight	−0.009	0.010	−0.078	−0.903	0.368
Propofol	−0.005	0.003	−0.162	−1.758	0.081
Midazolam	−0.006	0.005	−0.092	−1.132	0.260
Thiopental	−0.010	0.007	−0.125	−1.520	0.131
Sufentanil	−0.124	0.168	−0.064	−0.733	0.465
Piritramide	0.051	0.053	0.078	0.952	0.343
Fentanyl	−1.727	1.009	−0.141	−1.711	0.089
Arterial Hypertension	−0.108	0.336	−0.027	−0.321	0.749
Diabetes Mellitus	0.302	0.559	0.044	0.541	0.590
Hypothyroidism	−0.478	0.643	−0.061	−0.743	0.458
COPD	−0.083	0.856	−0.009	−0.097	0.923
Time after ITM	1.223 × 10^−5^	0.000	0.020	0.252	0.801

Remarks: N = 153; R^2^ = 0.154; corr. R^2^ = 0.068; F (14, 138) = 1.791; *p* = 0.045. b = regression coefficient; *SE* = standard error of the coefficient; *β* = standardised regression coefficient; *t* = *t*-value; *p* = *p*-value; R^2^ = coefficient of determination; F = F-statistic; ITM = intrathecal morphin; COPD = chronic obstructive pulmonary disease.

## Data Availability

Should further elucidation be required to support the findings of this study, the corresponding author is available to provide the necessary data upon reasonable request.

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
