# Peer review of "A Retrospective Cohort Study on the Side Effects of Intrathecal Morphine Administration Combined with General Anaesthesia Versus General Anaesthesia Alone in Prostatectomy Patients"

_medicina, 2025, doi:10.3390/medicina61040732_

Round 1
Reviewer 1 Report
Comments and Suggestions for Authors
The entire manuscript should be more concise. In particular:
- in the section dedicated to "methods" the authors should try to be more schematic in the description, so as to make the study repeatable by anyone who wants to do it. In this section it is not necessary to mention the limitations of the research, which will be done at the end of the "discussion".
- The "results" are described with an incredible (perhaps excessive) wealth of details. A greater schematization and simplification would help the reader to better understand what was found in the very in-depth retrospective study of the case history of the hospital being researched.
- The "discussion", as currently presented, is unacceptable. An appropriate "discussion" should begin with a very concise analysis of the main results obtained in the study. Immediately after these should be compared (discussed) with what has already been published on the subject, trying to clarify what the research being presented has added useful to what was already known. This "discussion", very long-winded, begins by describing things that should be in the "introduction". Then it further describes (even in a very repetitive way) all the tests done and the results obtained; something already done in the description of the "results". Lastly, finally, it dedicates a section to the "discussion", as it should be.
- The section dedicated to the limitations of the study, although it deserves a revision of the language, is definitely acceptable.
- The "conclusions" of a study of this kind do not need to be so (too) extensive. In the "conclusions" you should state, in 1-2 sentences, the results obtained. Then, in another 1-2 sentences, you should tell the reader what the research you intend to publish has added to what was already known, so as to make it useful from a clinical perspective.
Hopefully this will be useful to obtain a better product to re-submit.
Comments on the Quality of English LanguageThe scientific English of this manuscript should be revised by a native English-speaking professional medical writer.
Author Response
Dear Reviewer,
Thank you very much for your valuable feedback. We greatly appreciate your insights and have carefully revised the manuscript to address your suggestions. Below, we outline the specific changes we have made in response to your comments.
1. "In the section dedicated to 'methods' the authors should try to be more schematic in the description, so as to make the study repeatable by anyone who wants to do it. In this section it is not necessary to mention the limitations of the research, which will be done at the end of the 'discussion'."
- We have revised the Methods section to present the study in a more structured and schematic manner, ensuring greater clarity and reproducibility. Unnecessary details have been removed, and key methodological steps are now presented more concisely. Additionally, we have relocated the discussion of study limitations to the Discussionsection, as suggested, to maintain a clear distinction between methodology and critical evaluation.
2. "The 'results' are described with an incredible (perhaps excessive) wealth of details. A greater schematization and simplification would help the reader to better understand what was found in the very in-depth retrospective study of the case history of the hospital being researched."
- We have carefully revised the Results section to improve readability by reducing excessive details and structuring the findings more concisely. In particular, we have significantly shortened the description of the hemodynamic effects, removing redundant details while retaining the essential findings.
3. "The 'discussion', as currently presented, is unacceptable. An appropriate 'discussion' should begin with a very concise analysis of the main results obtained in the study. Immediately after these should be compared (discussed) with what has already been published on the subject, trying to clarify what the research being presented has added useful to what was already known. This 'discussion', very long-winded, begins by describing things that should be in the 'introduction'. Then it further describes (even in a very repetitive way) all the tests done and the results obtained; something already done in the description of the 'results'. Lastly, finally, it dedicates a section to the 'discussion', as it should be."
- We have thoroughly revised the Discussion section to align with standard scientific structure and improve clarity. Specifically, we have:
- Reorganized the section to begin with a concise summary of the main findings.
- Removed introductory content that was more appropriate for the Introduction.
- Eliminated redundant descriptions of statistical tests and results already covered in the Results section.
- Strengthened the comparison with existing literature, emphasizing how our findings contribute to the current understanding of the topic.
4. "The section dedicated to the limitations of the study, although it deserves a revision of the language, is definitely acceptable."
- We appreciate your positive assessment of the Limitations section and have carefully revised the language for improved clarity and readability. Additionally, we have ensured that the limitations are clearly structured and directly related to the study’s scope. We have emphasized key aspects such as the retrospective design, the single-center setting, and the potential for selection bias due to variations in clinical practice over the study period. Furthermore, we acknowledge the limitations of sample size imbalance between groups and the absence of long-term follow-up data.
5. "The 'conclusions' of a study of this kind do not need to be so (too) extensive. In the 'conclusions' you should state, in 1-2 sentences, the results obtained. Then, in another 1-2 sentences, you should tell the reader what the research you intend to publish has added to what was already known, so as to make it useful from a clinical perspective."
- We have revised the Conclusions section to be more concise by summarizing the main results in 1–2 sentences and clearly stating the clinical relevance in another 1–2 sentences. Redundant details have been removed to improve clarity and focus. We appreciate your suggestion, which has helped refine the manuscript.
We sincerely appreciate your thorough review and valuable suggestions, which have helped us enhance the clarity, structure, and overall quality of the manuscript.
Reviewer 2 Report
Comments and Suggestions for Authors
Dear authors,
Thank you for submitting your work. You are requested to revise the submission based on the following comments and also as per the comments in the PDF.
The results are extensively described. The authors could summarize the key findings, and the rest could be summarized in a table and/or figure with appropriate legends. Reading this much text is cumbersome for the readers also.
The number of patients is disproportionate in both groups. This is another limitation. Please mention that also.
Table 1 depicts the comparison of demography. However, the p-value is missing. Please provide that.
Table 2,3: provide expansion of the abbreviations used as footnote.
The complications part in the results is too detailed. Kindly mention the key findings and cite table and figures, as necessary.
The conclusions are very extensively written. It has almost the word count of the abstract. The conclusion should have a single paragraph, and should be summarized in 3-4, well constructed sentences.

Author Response
Dear Reviewer,
Thank you for your valuable feedback and for taking the time to review our manuscript. We appreciate your constructive comments and have carefully revised the submission accordingly. Below, we outline the specific changes made in response to your suggestions. Additionally, we have incorporated all modifications as per the comments provided in the PDF.
"The results are extensively described. The authors could summarize the key findings, and the rest could be summarized in a table and/or figure with appropriate legends. Reading this much text is cumbersome for the readers also."
- We have streamlined the Results section by summarizing the key findings concisely while moving additional details into tables and figures with appropriate legends. This revision enhances readability and improves the clarity of the findings.
"The number of patients is disproportionate in both groups. This is another limitation. Please mention that also."
- We have now explicitly mentioned the disproportionate group sizes as an additional limitation in the Limitationssection, as this could influence statistical power and interpretation of results.
"Table 1 depicts the comparison of demography. However, the p-value is missing. Please provide that."
- We have updated Table 1 to include the p-values for the demographic comparisons, ensuring transparency in the statistical evaluation.
"Table 2,3: provide expansion of the abbreviations used as footnote."
- We have revised Tables 2 and 3 to include a footnote with the expansion of abbreviations, making it easier for readers to understand the terms used.
"The complications part in the results is too detailed. Kindly mention the key findings and cite table and figures, as necessary."
- The Complications section has been condensed, highlighting only the key findings. Detailed descriptions have been removed, and we now refer to tables and figures where necessary for further information.
"The conclusions are very extensively written. It has almost the word count of the abstract. The conclusion should have a single paragraph, and should be summarized in 3-4, well-constructed sentences."
- We have significantly shortened the Conclusions section, reducing it to a single paragraph of 3–4 concise sentences, clearly summarizing the main findings and clinical implications.
We sincerely appreciate your insightful feedback, which has helped us improve the clarity and structure of our manuscript. Thank you for your time and consideration.
Round 2
Reviewer 1 Report
Comments and Suggestions for Authors
Thank you for the revision made. Now the submission is acceptable.
Reviewer 2 Report
Comments and Suggestions for Authors
Dear authors,
Thank you for revising your submission based on the comments raised. I am happy to see a more concise draft, with adequate flow, and results summarized succinctly in tables, to avoid an extensive read.
Best wishes.